# Skin Physiological Parameters and Their Association with Severe Atopic Dermatitis in Mongolian Children

**DOI:** 10.3390/jcm14010112

**Published:** 2024-12-28

**Authors:** Lkhamdari Batbileg, Sevjidmaa Baasanjav, Khosbayar Tulgaa, Oyuntugs Byambasukh, Khurelbaatar Naymdavaa, Enkhtur Yadamsuren, Baasanjargal Biziya

**Affiliations:** 1Department of Physiology, School of Biomedicine, Mongolian National University of Medical Sciences, Ulaanbaatar 14210, Mongolia; amd21f051@gt.mnums.edu.mn (L.B.);; 2Institute of Human Genetics, Martin Luther University Halle-Wittenberg, 06112 Halle (Saale), Germany; 3Clinical Molecular Diagnostic Center, School of Medicine, Mongolian National University of Medical Sciences, Ulaanbaatar 14210, Mongolia; khosbayar.t@mnums.edu.mn; 4Department of Clinical Laboratory, School of Medicine, Mongolian National University of Medical Sciences, Ulaanbaatar 14210, Mongolia; 5Department of Endocrinology, School of Medicine, Mongolian National University of Medical Sciences, Ulaanbaatar 14210, Mongolia; oyuntugs@mnums.edu.mn; 6Department of Dermatology, School of Medicine, Mongolian National University of Medical Sciences, Ulaanbaatar 14210, Mongolia

**Keywords:** epidermal barrier, skin physiology, capacitance, trans-epidermal water loss (TEWL), pH, climate, Mongolia

## Abstract

**Background**: Atopic dermatitis (AD) is a chronic skin condition that weakens the skin barrier, leading to increased trans-epidermal water loss and reduced skin moisture. Understanding how these changes in the skin barrier relate to AD severity in Mongolian children may offer insights that could apply to other regions facing similar environmental challenges. **Methods**: A cross-sectional study was conducted at the National Dermatology Center of Mongolia, involving 103 children with AD. Severity was assessed using the SCORAD index, and skin barrier function was measured through TEWL, skin moisture, and pH. Linear regression analyses were conducted, adjusting for age, skin physiological parameters, AD severity characteristics, and total IgE levels. **Results**: Among the participants, 48.54% were classified as having moderate AD, while 34.95% had severe AD. The mean SCORAD index was 43.19 ± 17.11. In the final adjusted regression analysis, higher TEWL was significantly associated with greater AD severity (non-lesional: B = 0.328, *p* = 0.004; lesional: B = 0.272, *p* = 0.007), while skin moisture showed an inverse association (non-lesional: B = −0.771, *p* < 0.001; lesional: B = −0.218, *p* < 0.001). The total IgE level was significantly higher in the severe AD group (*p* = 0.013). Although skin pH initially correlated with AD severity, it did not remain significant in multivariate analysis. **Conclusions**: This study emphasizes the role of skin barrier function, particularly increased TEWL and reduced moisture, in AD severity among Mongolian children.

## 1. Introduction

Atopic dermatitis (AD) is a chronic inflammatory skin disorder characterized by dry skin, eczematous lesions, and impaired skin barrier function. The prevalence of AD has surged globally, with a two- to threefold increase observed in developed countries over the past 30 years [1]. Furthermore, there is mounting evidence that allergic diseases, including AD, have also increased substantially in low- and middle-income countries [2]. In Mongolia, the number of diagnosed AD cases has risen significantly, from 11,672 in 2016 to 18,094 in 2019, with approximately 80% of these cases occurring in individuals under the age of 18 [3]. Similar trends are observed in other Asia–Pacific regions, where the prevalence of AD in early childhood has been reported to range from 7% to 27% [4,5,6,7,8].

The variability in AD prevalence and severity across different countries and regions has been well documented. For instance, the International Study of Asthma and Allergies in Childhood (ISAAC) Phase Three identified considerable variation in AD rates, influenced by environmental factors, socioeconomic conditions, and healthcare access [5,9]. In a large-scale study conducted across 18 countries, including the United States, Japan, and Russia, the prevalence of severe AD ranged from 2.2% to 18.1% of children [10]. These regional differences underscore the multifactorial nature of AD, where both intrinsic factors (such as genetic predisposition and immune dysregulation) and extrinsic factors (such as environmental conditions) play crucial roles. There is a limited amount of data available on intrinsic factors contributing to AD. However, Narmandakh et al. investigated the *FLG* gene in 46 Mongolian individuals with AD and identified six novel SNPs. However, they did not detect the 2281del and R501X mutations commonly observed in European populations [11]. Filaggrin is a key protein involved in the keratinization process critical for modulating the skin barrier and involving keratinocytes, and it exhibits genetic diversity and adaptation across different populations, including Asian populations [12]; additionally, Asian skin is reported to have a higher density of eccrine glands and a thinner stratum corneum, which may contribute to increased skin sensitivity to exogenous chemicals and affect barrier function [13]. Another factor that may contribute to the increased severity of AD in Mongolia is the low levels of 25(OH)D. This deficiency is likely due to the country’s high latitude, limited sun exposure during winter and spring, and restricted access to vitamin D-rich foods [14].

Although the specific impact of Mongolia’s extreme climatic conditions—marked by low humidity and significant seasonal fluctuations—on AD severity has not been directly studied, it is possible that these factors contribute to skin barrier impairment. The average relative humidity in Mongolia ranges from 20 to 30% during the spring and autumn months and drops below 20% in winter [15]. In contrast, other countries with milder climates, such as Korea and Japan, experience humidity levels between 40 and 80% [16,17,18]. Humidity is influenced by multiple factors including precipitation, temperature, air pressure, wind, geography, seasonal variation, vegetation, etc. In Mongolia, precipitation levels are particularly low, with values under 30 mm classified as dry. As shown in Figure 1, precipitation increases to 20–50 mm only for 1–3 days from June to August [19].

The lower humidity levels in Mongolia may exacerbate the skin’s dryness and contribute to AD severity by further compromising skin barrier function. Assessing the integrity of the skin barrier is critical for understanding AD pathophysiology. Key markers of skin barrier function include trans-epidermal water loss (TEWL) and skin moisture, both of which provide insight into the skin’s ability to retain water and protect against external irritants [20,21,22]. Higher TEWL and lower skin moisture are typically indicative of a compromised skin barrier, a hallmark of AD. Additionally, skin pH plays an important role in maintaining skin homeostasis, as elevated pH levels can lead to increased TEWL and further barrier disruption [23,24].

In this study, we aim to assess the severity of AD in Mongolian children and explore the relationship between skin physiological parameters—TEWL, skin moisture, and skin pH—and AD severity. Given the high burden of AD in Mongolia and the potential influence of environmental factors on skin barrier function, understanding these relationships may provide new insights into the management of AD in regions with challenging climates.

## 2. Materials and Methods

### 2.1. Data Collection and Study Participants

A cross-sectional study was conducted from April to May 2023 at the National Dermatology Center of Mongolia, involving 103 children under the age of 18 diagnosed with AD according to the Hanifin and Rajka criteria [25]. Exclusion criteria included the presence of other skin diseases, allergic diseases, chronic conditions such as diabetes, liver or renal impairment, and the use of phototherapy or systemic therapies that could impact the study parameters. Participants were asked to avoid using moisturizers for at least 24 h before the study and to stop using topical steroid treatments for at least one week.

Informed consent was obtained from all participants, and the study was approved by the Ethics Committee of the Mongolian National University of Medical Sciences (Approval No. 2022/3-05).

### 2.2. Atopic Dermatitis Severity

The severity of AD was determined using the SCORAD (Scoring Atopic Dermatitis) index, which combines both objective and subjective criteria. Objective criteria include the extent of skin involvement (percentage of body surface affected) and the severity of six symptoms: erythema, edema, oozing/crusting, excoriation, lichenification, and dryness, each graded from 0 to 3. Subjective measures, including pruritus and sleep disturbance, were rated on a scale of 0 to 10. These components together provide a total SCORAD score ranging from 0 to 103, with higher scores indicating greater disease severity. Based on the total score, participants were classified into three categories: mild (<25 points), moderate (25–50 points), and severe (>50 points) [26].

### 2.3. Skin Parameters

Skin physiological parameters, including moisture content, TEWL, and skin pH, were measured following guidelines provided by the European Group on Efficacy Measurement and Evaluation (EEMCO). The MC750 device (Courage+Khazaka electronic GmbH, Köln, Germany) was used for these assessments [27,28,29]. Skin moisture was measured with a Corneometer^®^ CM 825 probe (Courage+Khazaka electronic GmbH, Köln, Germany), pH was evaluated with the Skin-pH-Meter PH 905 (Courage+Khazaka electronic GmbH, Köln, Germany), and TEWL was assessed using the Tewameter^®^ TM Hex (Courage+Khazaka electronic GmbH, Köln, Germany).

All measurements were taken at nine different body sites which were not affected by dermatitis, including the forehead, cheek, antecubital region, volar and dorsal forearm, abdomen, interscapular region, popliteal fossa, and the posterior leg [30]. This measurement is referred to as the “non-lesional area”, while measurements from skin sites with eczematous lesions were analyzed separately, under the name “lesional area”. Each site was measured three times, and the mean value was recorded. Before taking measurements, participants were allowed to acclimatize for 20 min in a controlled environment, with humidity maintained between 40 and 45% and temperature between 20 and 22 °C [27,28,29].

### 2.4. Other Parameters

Information on participants’ general characteristics—such as age, breastfeeding status, and birth weight—along with details regarding their AD history, including age of onset, duration of the condition, history of hospitalizations, and school or preschool absences, was gathered using a modified version of the International Study of Asthma and Allergies in Childhood (ISAAC) questionnaire [5].

Serum total IgE levels were measured using the AccuBind ELISA Microwells system (Monobind LLC, Lake Forest, CA, USA), employing the sandwich ELISA technique. The analysis was performed at the clinical laboratory of the National Dermatology Center.

### 2.5. Statistical Analysis

Data were summarized using descriptive statistics: continuous variables were presented as mean ± standard deviation for normally distributed data, or as medians (with minimum and maximum values) for non-normally distributed data. Categorical variables were summarized using frequencies and percentages. The normality of distributions was tested with histograms and the Skewness–Kurtosis test. SCORAD scores and skin parameters in the non-lesional area followed normal distributions, while age, age of onset, skin parameters in the lesional area, and total IgE levels were non-normally distributed.

To compare differences between groups, ANOVA was used for normally distributed continuous variables, the Kruskal–Wallis test was applied for non-parametric data, and the Chi-square test was used for categorical variables. Pearson’s correlation was used for parametric data to assess correlations between variables, while Spearman’s rank correlation was applied for non-parametric variables.

Linear regression models were used to assess associations between skin physiological parameters and AD severity (measured by the SCORAD index), with adjustments applied progressively. Model 1 adjusted for age; Model 2A included age and additional skin physiological parameters; Model 2B incorporated age along with general characteristics, including age of AD onset, hospital admission, and school absence; Model 3 included age, skin physiological parameters, and general characteristics; Model 4 included age, skin physiological parameters, general characteristics, recent moisturizer application history (last 6 months), and daily moisturizing habits; Model 5 included age, skin physiological parameters, general characteristics, and living conditions, including housing type, living area, and regional distribution; and Model 6 further included age and total IgE levels. In the regression analysis, the independent variables—skin moisture, TEWL, and pH—from the lesional areas were transformed using the natural logarithm to achieve a normal distribution. Although the parameters from the non-lesional areas and SCORAD index were normally distributed, they were also log-transformed to ensure consistency and facilitate comparison between lesional and non-lesional areas. Statistical analyses were performed using Stata-14 software (StataCorp LLC, College Station, TX, USA), and a *p*-value of <0.05 was considered statistically significant.

## 3. Results

A total of 103 participants with AD were included, stratified into mild (n = 17), moderate (n = 50), and severe (n = 36) groups based on SCORAD index scores. The mean SCORAD index was 43.19 ± 17.11, indicating a range of disease severities. Figure 2 illustrates the SCORAD distribution, with thresholds for mild, moderate, and severe AD. Figure 3 shows the lesional images of participants with severe atopic dermatitis.

Table 1 summarizes the demographic and clinical characteristics. The median age was 5 years, with no significant age difference across severity groups (*p* = 0.991). Gender distribution showed 41.7% male participants, with no significant variation between groups (*p* = 0.693). The age of AD onset differed significantly (*p* = 0.027), with an earlier median onset in the severe group, suggesting a potential link between earlier onset and increased severity. However, AD duration did not vary significantly across groups (*p* = 0.526). Breastfeeding rates were high across all groups (93.2%, *p* = 0.670), and birth weight distributions were similar (*p* = 0.915). Notably, hospital admissions were significantly more frequent in the severe AD group (44.4%, *p* < 0.001), as was school absenteeism due to AD (66.7% in severe cases, *p* = 0.015), emphasizing the substantial impact of AD severity on healthcare use and daily life. Housing type and living area showed no significant differences between severity groups, with the majority of participants residing in apartments (71.57%, *p* = 0.792) and urban areas (73.29%, *p* = 679). Additionally, regional distribution was comparable across groups, with no significant differences (*p* = 0.729). Recent moisturizer application history was consistent across all groups (90.29%) but did not significantly differ between them (*p* = 0.300). However, daily moisturizing habits were significantly less frequent in the mild group (35.29%) compared to the moderate (54.00%) and severe groups (55.56%, *p* = 0.020). Total IgE levels were significantly higher in participants with severe AD (*p* = 0.013), with median levels of 229.45 IU/mL, aligning with established evidence of immune dysregulation in AD pathophysiology. Higher IgE levels in severe AD suggest heightened immune activation associated with increased disease severity.

Table 2 summarizes the skin physiological parameters across AD severity groups. In non-lesional areas, skin moisture levels showed a significant decrease with increasing AD severity, indicating drier skin in more severe cases (*p* < 0.001). Although TEWL tended to increase with AD severity, this difference was not statistically significant (*p* = 0.138). Skin pH was significantly higher in participants with severe AD compared to those with mild and moderate AD (*p* = 0.049). In lesional areas, skin moisture levels were significantly lower in the severe AD group, reflecting greater moisture loss (*p* = 0.002). TEWL was notably higher in severe cases, suggesting compromised barrier function in affected skin areas (*p* < 0.001). However, no significant differences were observed in skin pH across severity groups in lesional areas (*p* = 0.185).

The SCORAD index showed significant associations with skin physiological parameters (Figure 4). In non-lesional areas, TEWL positively correlated with SCORAD scores (r = 0.267, *p* = 0.006), indicating increased water loss in severe AD cases. Skin moisture had a significant negative correlation (r = −0.456, *p* < 0.001), and skin pH showed a weaker positive correlation (r = 0.277, *p* = 0.005). Similar correlations were observed in lesional areas, with negative associations for skin moisture (r = −0.335, *p* < 0.001) and positive associations for TEWL (r = 0.370, *p* < 0.001), while skin pH showed no significant association (r = 0.128, *p* = 0.196).

The regression analyses, as shown in Table 3 and Table 4, highlight the associations between AD severity (measured by the SCORAD index) and skin physiological parameters. In the univariate analysis, TEWL and skin moisture emerged as strong predictors of AD severity. Specifically, increased TEWL was associated with higher SCORAD scores, while reduced skin moisture was linked to greater disease severity. The relationship between skin pH and SCORAD index, while significant in the univariate analysis, did not retain significance in the multivariate models, indicating that its effect may be influenced by other skin barrier functions.

In the multivariate regression analyses, several models were constructed to adjust for various confounders while examining the influence of skin physiological parameters on the SCORAD index in both lesional and non-lesional skin areas. These models controlled for age, age of AD onset, hospital admissions, school absences, and living environment. Despite these comprehensive adjustments, TEWL and skin moisture consistently emerged as significant predictors of AD severity in both lesional and non-lesional areas, highlighting their fundamental roles in the progression of the disease. Further adjustments in Model 6, which also included total IgE levels, demonstrated that TEWL and skin moisture maintained their significant associations with AD severity in both skin areas. This persistence underscores the critical role of skin barrier dysfunction in AD, independent of immune response markers, suggesting that interventions aimed at enhancing skin barrier function may be effective regardless of underlying immunological factors. These robust associations have been documented in Table 4A,B.

## 4. Discussion

This study explored the relationship between skin physiological parameters (TEWL, moisture, and pH) and AD severity in Mongolian children, contributing valuable insights into how skin barrier function correlates with the progression of AD. With nearly half of the participants categorized as having moderate AD (48.54%) and more than a third (34.95%) presenting severe AD, the findings reflect a significant disease burden in this pediatric population. The mean SCORAD index of 43.19 ± 17.11 is notably higher than that reported in other geographic regions, where AD tends to present with lower SCORAD scores ranging from 22.13 ± 22.00 to 40.8 ± 21.48 [20,31,32,33,34,35,36]. In the study by Ulzii et al. on the severity strata of Patient-Oriented Eczema Measure (POEM) scores in patients with atopic dermatitis in Mongolia, the average POEM score was 16.58 ± 7.33, indicating moderate severity. Although this study used different severity scoring methods, the average result consistently showed moderate severity [37]. These results may highlight a more severe manifestation of AD in Mongolian children, possibly influenced by factors not fully addressed in this study, such as environmental conditions and access to healthcare. Low temperatures have been notably linked to a significant increase in outpatient visits for AD, with children being the most affected [38]. Furthermore, low vapor pressure and humidity levels are strongly correlated with a higher incidence of AD, suggesting that maintaining adequate humidity levels could be beneficial in managing the condition [39]. As highlighted in the introduction, Mongolia experiences low humidity levels and harsh temperatures, which may exacerbate AD symptoms by impairing skin barrier function. Our study also observed an early age of onset for AD, with a median onset age of 0 years to 1 year across all severity groups, likely reflecting intrinsic genetic factors [40]. Although the specific role of climate and genetic factors in AD severity was not the focus of this research, the consistently higher SCORAD values observed here suggest a possible interaction between skin barrier dysfunction and external factors such as humidity and temperature, warranting further exploration.

Our results demonstrated significant changes in skin physiological parameters as AD severity increased. TEWL, an indicator of skin barrier function, was significantly elevated in participants with more severe AD (B = 0.328, *p* = 0.004), while skin moisture, a measure of hydration in the stratum corneum, decreased as disease severity worsened (B = −0.771, *p* < 0.001). These findings confirm that a compromised skin barrier is closely linked to AD progression and suggest that targeting skin hydration and reducing water loss are essential components of managing the condition. Higher TEWL indicates increased permeability of the skin, which facilitates water loss and allows irritants and allergens to penetrate more easily, aggravating the inflammatory response characteristic of AD [20,22,24,30]. The significant decrease in skin moisture further emphasizes the importance of maintaining proper hydration in managing AD, particularly in regions where environmental conditions may exacerbate skin dryness.

In contrast, skin pH, which regulates several physiological processes and is known to influence the skin’s susceptibility to irritation and infection, did not remain a significant predictor of AD severity in our multivariate analysis. While elevated pH is often associated with increased TEWL and impaired barrier function, in this study, it did not demonstrate the same level of importance as TEWL and moisture in determining AD severity [22,24,30,41]. This may be due to the fact that pH, while important, is one of several factors that interact to affect skin barrier integrity. Moreover, the variability in pH measurement techniques and the influence of extrinsic factors (such as soap use and topical treatments) may complicate its direct relationship with AD severity [42,43]. Thus, further research is needed to evaluate the role of pH in AD progression, especially in populations exposed to different environmental conditions.

Our study also revealed that immune hypersensitivity, as measured by total IgE levels, plays a significant role in the severity of AD. Elevated total IgE levels were significantly correlated with increased SCORAD scores (*p* < 0.05), indicating that heightened immune responses exacerbate AD symptoms. Approximately 43.69% of participants exhibited elevated IgE levels, suggesting that immune activation is a critical driver of AD severity in this population. These findings align with the well-established role of IgE in the pathophysiology of AD, where immune dysregulation contributes to the chronic inflammation seen in more severe cases [44,45,46]. The significant relationship between IgE and the SCORAD index suggests that therapeutic approaches targeting both skin barrier repair and immune modulation could offer a comprehensive strategy for managing severe AD in children.

This study has several strengths. The data were collected at a national dermatology center, the sole nationwide open-access dermatology facility which may be representative of the atopic dermatitis cases treated in Mongolian healthcare settings. This center likely treats a broad cross-section of the pediatric population with AD, enhancing the generalizability of our findings to other hospitals and regions in Mongolia. Additionally, the use of standardized objective measures, such as TEWL, skin moisture, and pH, ensures that the assessment of skin barrier function was conducted with high accuracy and reliability, allowing for meaningful comparisons with international studies. However, there are limitations to consider. First, as a cross-sectional study, it is not possible to establish causality between skin barrier dysfunction and AD severity. Longitudinal studies would be required to explore how skin physiological parameters evolve over time and in response to treatment. Additionally, this study did not directly assess the influence of environmental factors, such as seasonal variations in humidity and temperature, on AD severity. Therefore, we did not analyze data across all four seasons for comparison, which is one of the limitations of this study; however, it is important to note that Mongolia has a dry and extreme climate throughout the year. Given Mongolia’s unique climate, future research should examine how these external factors contribute to skin barrier impairment and disease progression. Another limitation is the gender imbalance in the sample, as the majority of participants were female. A more balanced sample would improve the generalizability of the findings across genders. Despite these limitations, our study provides valuable insights into the physiological aspects of AD in Mongolian children, particularly the role of skin barrier dysfunction as reflected by increased TEWL and decreased skin moisture. The findings suggest that interventions targeting skin barrier repair could play a critical role in managing AD severity, particularly in regions with harsh environmental conditions like Mongolia. Finally, our study contributes important information that could benefit researchers and clinicians, particularly those working in climatically challenging or underserved areas.

## 5. Conclusions

This study highlights the importance of skin barrier function, particularly increased TEWL and reduced skin moisture, in relation to AD severity in Mongolian children. These associations persisted even after adjusting for age, AD severity characteristics, and immune markers, underscoring the role of skin hydration and integrity in AD management. While skin pH was initially related to severity, it did not remain significant in adjusted models, suggesting that TEWL and moisture are more reliable indicators. These findings may inform strategies to support skin barrier health in populations vulnerable to AD, especially where environmental conditions could exacerbate barrier dysfunction.

## Figures and Tables

**Figure 1 jcm-14-00112-f001:**
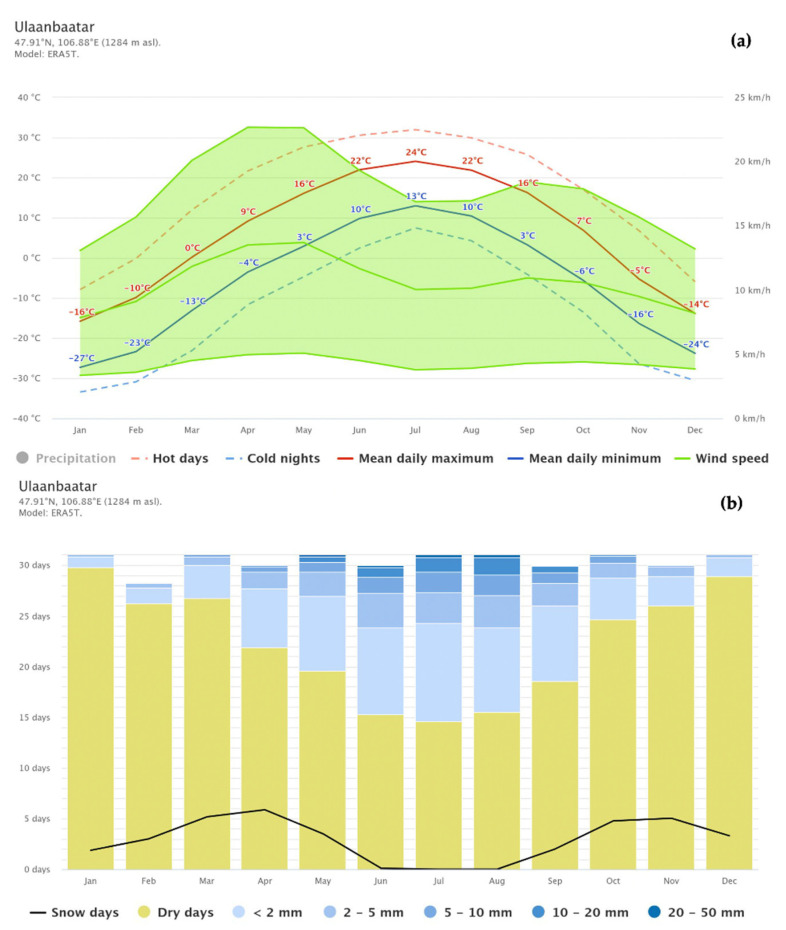
(**a**) The mean daily maximum (red line) and minimum (blue line) temperatures represent the average maximum and minimum temperatures for each month in Ulaanbaatar over the past 30 years. Wind speeds are represented by the green line and the green shaded area. The dashed blue line indicates the threshold for cold nights, while the dashed red line marks the threshold for hot days [19]. (**b**) Monthly precipitation levels are categorized as mostly wet when exceeding 150 mm and mostly dry when below 30 mm. The graph depicts the average daily precipitation levels for each month in Ulaanbaatar. The black line shows the number of snowy days per month [19].

**Figure 2 jcm-14-00112-f002:**
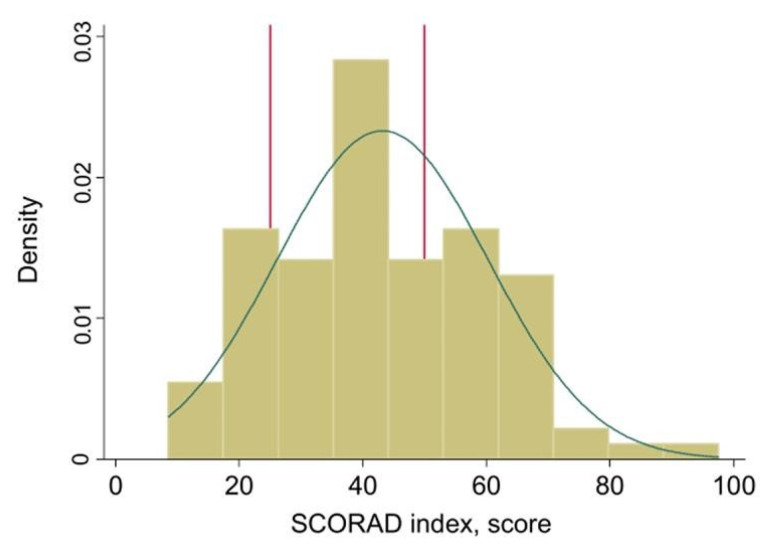
Histogram and density curve of SCORAD index scores with reference lines marking thresholds for mild (<25), moderate (25–50), and severe (>50) atopic dermatitis.

**Figure 3 jcm-14-00112-f003:**
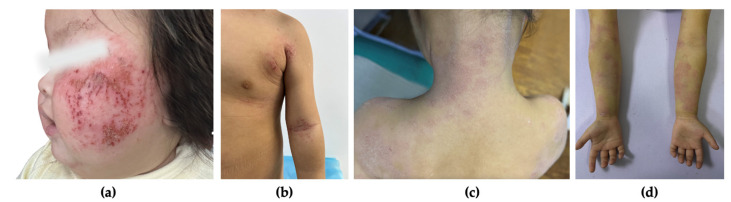
Lesional images of participants with severe atopic dermatitis. (**a**) Erythema, excoriations, and crusting localized on the cheek in a 6-month-old infant. (**b**) Erythema, excoriations, and crusting localized on the arm and torso in a 3-year-old child. (**c**) Erythema, scaling, and lichenification observed on the neck and upper back in a 5-year-old child. (**d**) Erythema, scaling, and lichenification observed on the forearms in a 7-year-old child.

**Figure 4 jcm-14-00112-f004:**
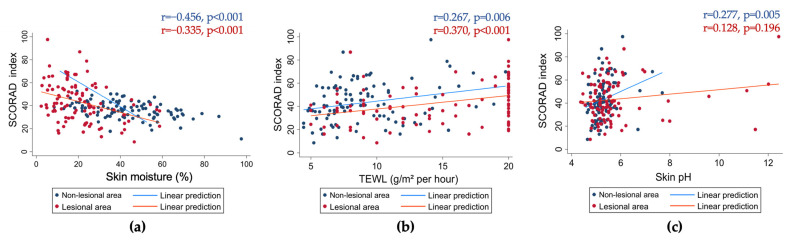
The correlation between SCORAD index and skin physiological parameters. (**a**) Skin moisture. (**b**) TEWL. (**c**) Skin pH. Measurements from the non-lesional areas are represented in navy, while measurements from the lesional areas are shown in red. Pearson correlation was used for non-lesional areas, while Spearman correlation was used for lesional areas. A visual representation of these correlations. Panel (**a**) illustrates the negative correlation between skin moisture and the SCORAD index, showing that lower moisture levels are associated with more severe AD. Panel (**b**) shows a positive correlation between TEWL and SCORAD index. Panel (**c**) demonstrates a weak but significant positive correlation between skin pH and SCORAD index.

**Table 1 jcm-14-00112-t001:** Characteristics of study participants.

Parameters	Total	AD Severity Groups			*p*-Value
		Mild (n = 17)	Moderate (n = 50)	Severe (n = 36)	
Age, years	5 [0.25–18]	6 [0.33–18]	5 [0.5–18]	5 [0.25–18]	0.991 **
Gender, % (n)					
Male	41.70 (43)	41.18 (7)	38.00 (19)	47.22 (17)	0.693 ***
Female	58.25 (60)	58.82 (10)	62.00 (31)	52.78 (19)	
Age of AD onset, years	1 [0–17]	0 [0–4]	1 [0–17]	0 [0–13]	0.027 **
AD duration, years	3 [0–18]	4 [0–18]	3 [0–13]	4 [0–18]	0.526 **
Breastfeeding, % (n)	93.20 (96)	88.24 (15)	94.00 (47)	94.44 (34)	0.670 ***
Birth weight, gram					
<2500	8.74 (9)	11.76 (2)	10.00 (5)	5.56 (2)	0.915 ***
2500–3500	40.78 (42)	41.18 (7)	38.00 (19)	44.44 (16)	
>3500	50.49 (52)	47.07 (8)	52.00 (26)	50.00 (18)	
Housing type: apartment, % (n)	71.57 (73)	76.47 (13)	67.35 (33)	75 (27)	0.792 ***
Living area: urban, % (n)	73.29 (76)	82.35 (14)	72.00 (36)	72.22 (26)	0.679 ***
Regional distribution, % (n)					
Eastern	29.13 (30)	35.29 (6)	30.00 (14)	25.00 (9)	0.729 ***
Central	42.72 (44)	29.41 (5)	42.00 (21)	50.00 (18)	
Western	28.16 (29)	35.29 (6)	28.00 (14)	25.00 (9)	
Recent moisturizer application history (last 6 months), % (n)	90.29 (93)	100 (17)	88.00 (44)	88.89 (32)	0.300 ***
Daily moisturizing, % (n)	51.46 (53)	35.29 (6)	54.00 (27)	55.56 (20)	0.020 ***
Hospital admission, % (n)	21.36 (22)	5.88 (1)	10 (5)	44.44 (16)	<0.001 ***
School absence, % (n)	46.60 (48)	17.65 (3)	42.00 (21)	66.67 (24)	0.015 ***
Total IgE level, IU/ml	118.8 [2.3–458]	60 [7.1–337.5]	116.2 [2.3–436.4]	229.45 [6.1–458]	0.013 **

Notes. The data are reported as mean ± SD, median [minimum, maximum], and percentages (numbers). ** Kruskal–Wallis test was performed for non-parametric variables; *** Chi-square test was performed for categorical variables.

**Table 2 jcm-14-00112-t002:** Skin physiological parameters in non-lesional and lesional areas.

Parameters	Total	AD Severity Groups			*p*-Value
		Mild (n = 17)	Moderate (n = 50)	Severe (n = 36)	
Non-lesional area					
Skin moisture, %	36.27 ± 7.25	40.14 ± 5.24	37.59 ± 6.78	32.62 ± 7.28	<0.001 *
TEWL, g/m^2^/h	9.01 ± 3.44	8.08 ± 3.31	8.71 ± 3.35	9.89 ± 3.54	0.138 *
Skin pH	5.29 ± 0.49	5.13 ± 0.44	5.24 ± 0.46	5.44 ± 0.53	0.049 *
Lesional area					
Skin moisture, %	18.67 [2.33–58.67]	26 [10.33–58.67]	18.67 [2.33–58.67]	15.17 [2.67–33.33]	0.002 **
TEWL, g/m^2^/h	17 [5–20]	11 [5–20]	12.5 [5–20]	20 [7–20]	<0.001 **
Skin pH	5.42 [4.37–12.43]	5.36 [4.76–11.48]	5.31 [4.37–9.59]	5.50 [4.45–12.43]	0.185 **

Notes. The data are reported as mean ± SD, median [minimum, maximum], and percentages (numbers). * A one-way ANOVA test was performed for parametric variables; ** Kruskal–Wallis test was performed for non-parametric variables.

**Table 3 jcm-14-00112-t003:** Regression analysis of the SCORAD index in relation to skin physiological parameters.

Parameters	Univariate Regression			Multivariate Regression		
	B Coefficient	95% CI	*p* Value	B Coefficient	95% CI	*p* Value
Non-lesional area						
Skin moisture, %	−0.772	−1.116; −0.428	<0.001	−0.725	−1.068; −0.381	<0.001
TEWL, g/m^2^/h	0.327	0.092; 0.561	0.007	0.350	0.139; 0.561	0.001
Skin pH	1.415	0.442; 2.389	0.005	0.733	−0.190; 1.657	0.118
Lesional area						
Skin moisture, %	−0.226	−0.347; −0.106	<0.001	−0.168	−0.295; −0.040	0.011
TEWL, g/m^2^/h	0.329	0.137; 0.523	0.001	0.232	0.029; 0.435	0.025
Skin pH	0.230	−0.215; 0.676	0.308	0.2086	−0.205; 0.622	0.320

Notes. Regression analysis was conducted to assess AD severity groups in relation to the evaluation of skin barrier parameters. The results are presented as unstandardized beta coefficients with corresponding 95% confidence intervals (95% CIs).

**Table 4 jcm-14-00112-t004:** (**A**) Adjusted analysis of the SCORAD index with skin physiological parameters of the non-lesional area. (**B**) Adjusted analysis of the SCORAD index with skin physiological parameters of the lesional area.

(**A**)			
**Parameters**	**SCORAD Index-Linear Regression**		
	**B Coefficient**	**95% CI**	***p* Value**
TEWL			
Model 1	0.375	0.130; 0.621	0.003
Model 2A	0.341	0.119; 0.564	0.003
Model 2B	0.248	0.027; 0.469	0.028
Model 3	0.260	0.048; 0.473	0.017
Model 4	0.249	0.038; 0.462	0.022
Model 5	0.263	0.046; 0.482	0.018
Model 6	0.328	0.104; 0.551	0.004
Skin moisture			
Model 1	−0.857	−1.224; −0.490	<0.001
Model 2A	−0.741	−1.110; −0.373	<0.001
Model 2B	−0.532	−0.887; −0.177	0.004
Model 3	−0.523	−0.880; −0.166	0.005
Model 4	−0.515	−0.874; −0.158	0.005
Model 5	−0.513	−0.881; −0.144	0.007
Model 6	−0.771	−1.133; −0.411	<0.001
Skin pH			
Model 1	1.410	0.431; 2.390	0.005
Model 2A	0.727	−0.203; 1.656	0.124
Model 2B	0.572	−0.332; 1.477	0.212
Model 3	0.277	−0.600; 1.155	0.531
Model 4	0.234	−0.637; 1.104	0.596
Model 5	0.290	0.604; 1.184	0.521
Model 6	0.984	−0.031; 1.999	0.057
**(B)**			
**Parameters**	**SCORAD Index-Linear Regression**		
	**B Coefficient**	**95% CI**	***p* Value**
TEWL			
Model 1	0.334	0.140; 0.528	0.001
Model 2A	0.231	0.025; 0.437	0.028
Model 2B	0.209	0.031; 0.386	0.021
Model 3	0.126	−0.061; 0.313	0.185
Model 4	0.118	−0.069; 0.304	0.214
Model 5	0.129	−0.062; 0.321	0.183
Model 6	0.272	0.077; 0.467	0.007
Skin moisture			
Model 1	−0.228	−0.351; −0.106	<0.001
Model 2A	−0.169	−0.300; −0.037	0.012
Model 2B	−0.167	−0.275; −0.059	0.003
Model 3	−0.138	−0.254; −0.022	0.020
Model 4	−0.140	−0.257; −0.023	0.020
Model 5	−0.139	−0.258; −0.019	0.023
Model 6	−0.218	−0.335; −0.100	<0.001
Skin pH			
Model 1	0.223	−0.232; 0.678	0.334
Model 2A	0.211	−0.210; 0.632	0.323
Model 2B	−0.104	−0.507; 0.299	0.610
Model 3	−0.080	−0.467; 0.306	0.681
Model 4	−0.077	−0.463; 0.309	0.693
Model 5	−0.086	−0.482; 0.309	0.666
Model 6	0.111	−0.331; 0.553	0.618

Notes. Model 1 adjusted for age. Model 2A adjusted for age and additional skin physiological parameters. Model 2B adjusted for age and general characteristics, including age of AD onset, hospital admission, and school absence. Model 3 adjusted for age, skin physiological parameters, and general characteristics. Model 4 adjusted for age, skin physiological parameters, general characteristics, recent moisturizer application history (last 6 months), and daily moisturizing habits. Model 5 adjusted for age, skin physiological parameters, general characteristics, and living conditions, including housing type, living area, and regional distribution. Model 6 adjusted for age and total IgE levels.

## Data Availability

The data used to support the findings of this study are available from the corresponding author upon request.

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
