# Peer review of "Skin Physiological Parameters and Their Association with Severe Atopic Dermatitis in Mongolian Children"

_jcm, 2024, doi:10.3390/jcm14010112_

Round 1
Reviewer 1 Report
Comments and Suggestions for Authors
This is a manuscript of limited interest to the reader, but very well-written article, giving insights about
1. confounding factors, such as persoral history of atopic march, use of emollients, co-existing dermatoses or even socioeconomic status, could all influence the presentation of AD and should be addressed by the authors
2. Mongolia is diverse, both geographically and culturally, which is the reason why different ethnic groups / regions / urbanization levels should be taken into account
3. Another issue that should have been taken into account is mongolian skin itself and the inherent differences in baseline TEWL, skin moisture, and pH due to genetic factors.
Author Response
Point-by-point response to the reviewer comments
Manuscript: jcm-3337268
Reviewer 1.
This is a manuscript of limited interest to the reader, but very well-written article, giving insights about
Comment 1: Confounding factors, such as persoral history of atopic march, use of emollients, co-existing dermatoses or even socioeconomic status, could all influence the presentation of AD and should be addressed by the authors
Response: We appreciate the reviewer’s insightful comment. We have added a model to the regression analysis to account for these potential confounding factors, including socioeconomic status, and other relevant variables, ensuring a more comprehensive assessment of their impact on the presentation of AD (Table 4A, 4B). Socioeconomic status was evaluated using housing data and living area from the questionnaire, which is summarized in Table 1. Regarding other confounding variables, children were included in the study in accordance with EEMCO guidelines, with the requirement that no moisturizers were applied within the 24 hours preceding participation. Individuals with other skin conditions or atopic disorders were excluded, a detail further clarified in the first paragraph of the methodology section (Line 111).
Comment 2: Mongolia is diverse, both geographically and culturally, which is the reason why different ethnic groups / regions / urbanization levels should be taken into account
Response: Thank you for raising this point. This cross-sectional study was conducted at the National Dermatology Center in Ulaanbaatar, Mongolia, the country's only nationwide dermatology facility with open access, without any regional emphasis (Line 111). Indeed, we agree with the reviewer on the potential value of considering regional differences in our analysis. However, we did not observe a significant effect, which may be attributed to the uniformly harsh climatic conditions across Mongolia. For instance, desert areas with their low humidity and extreme temperature variations can exacerbate skin dryness and irritation, while the cold and windy conditions in western regions can lead to increased skin barrier disruption and dryness, aggravating AD severity. These uniformly challenging conditions might explain the lack of significant regional effects in our findings. We have included this observation in the results section by adding a model that incorporates these factors and shows no changes in the regression models (Table 4). Additionally, we have discussed this aspect in the discussion section to provide further context to the analysis results (Lines 372).
Comment 3: Another issue that should have been taken into account is Mongolian skin itself and the inherent differences in baseline TEWL, skin moisture, and pH due to genetic factors.
Response: We appreciate this thoughtful comment. In the revised version, we have included an introduction discussing the genetic specificity of skin (Lines 59-65). Additionally, we have used the physiological parameters from the participants' non-lesional area as the inherent baseline, as shown in Table 4A.
We appreciate the reviewers' valuable comments, which have helped improve the clarity and robustness of our manuscript.
Reviewer 2 Report
Comments and Suggestions for Authors
i read with great interest the manuscript on atopic dermatitis on mongolian children:
comments
- abstarct : well written with all the necessary info needed
- intro
exposome of atopic dermatitis is indeed a very challenging topic- the authors report very nicely the environmental factors affecting the course of the disease- are also some other individual-patient factors (inner exposome- genetics, skin composure characteristics) charactersing the mongolian skin ( organisition and size of keratinocytes in Asian populations, function of glands, pigmentaition) those should be briefly reported
methods are presented nicely- nothing to add
in results section correlations, tables and data presentations are presently very clearly well done maybe a image of lesions in mongolian patients with atopic dermatitis would improve to the manuscript
discussion
you report:The mean SCORAD index of 43.19 ± 17.11 is notably higher than that 248 reported in other geographic regions, ( can you name the factors that you believe affect those scores - sun exposure, dry climate,genes, skin of colour thatt make diagnosis harderer?? etc)
a table or a map or further report on text with examples of ethnicities and the mean SCORAD report comparsion with the Mongolian population would also increase the interest for the readers
Generally the manuscript is well presented and well written
Author Response
Point-by-point response to the reviewer comments
Manuscript: jcm-3337268
Reviewer 2.
Abstract: well written with all the necessary info needed
Response: We appreciate the reviewer’s positive feedback regarding the abstract.
Introduction: Exposome of atopic dermatitis is indeed a very challenging topic- the authors report very nicely the environmental factors affecting the course of the disease- are also some other individual-patient factors (inner exposome- genetics, skin composure characteristics) charactersing the mongolian skin ( organisition and size of keratinocytes in Asian populations, function of glands, pigmentation) those should be briefly reported
Response: Thank you for the insightful comment regarding the inner exposome. Atopic dermatitis (AD) is indeed a complex condition influenced by various environmental, genetic, and physiological factors. We agree that discussing patient-specific factors such as genetics, skin composition, keratinocyte organization, gland function, and pigmentation could enhance the study's comprehensiveness. To address this, we have added a brief discussion on these aspects in the Introduction, particularly emphasizing the unique characteristics of skin and highlighting the need for further research into these factors (Line 59).
Methods: Methods are presented nicely- nothing to add.
Response: We appreciate the reviewer’s positive evaluation of the methods section.
Results: In results section correlations, tables and data presentations are presently very clearly well done maybe a image of lesions in Mongolian patients with atopic dermatitis would improve to the manuscript
Response: We thank the reviewer for the suggestion to include images of lesions in Mongolian patients with atopic dermatitis. In response to your recommendation, we have added images of Mongolian children with AD to enhance the manuscript and provide additional context for the readers (Figure 3).
Discussion: You report: The mean SCORAD index of 43.19 ± 17.11 is notably higher than that 248 reported in other geographic regions, (can you name the factors that you believe affect those scores - sun exposure, dry climate, genes, skin of colour thatt make diagnosis harderer?? etc)
a table or a map or further report on text with examples of ethnicities and the mean SCORAD report comparison with the Mongolian population would also increase the interest for the readers
Response: We appreciate the reviewer’s comment on factors influencing the higher mean SCORAD index observed in our study. Based on your suggestion, we have expanded the discussion to include possible contributing factors, such as sun exposure, dry and harsh climate, genetic predispositions which may complicate diagnosis and exacerbate AD severity. Additionally, we have enriched the manuscript by adding text highlighting the unique challenges faced by the Mongolian population (Line 315).
Generally, the manuscript is well presented and well written.
Response: We appreciate the reviewer’s valuable suggestions, which have significantly improved the clarity and depth of our manuscript. All comments have been addressed, and the revised version reflects these changes.
Reviewer 3 Report
Comments and Suggestions for Authors
Atopic dermatitis (AD) is a chronic, recurrent dermatitis with periods of exacerbations and remissions. The disease is manifested by persistent itching, dryness and skin lichenification, what significantly contribute to the decrease of the quality of life of patients and their families. It is a widely known, common condition that usually appears in childhood, but adults also get sick. So, the topic is important.
In times when the way of life, climate and the amount of pollution are intensively changing, environmental factors play an important role in the severity of AD symptoms. The most common include: air pollution, exposure to stress, UV, certain nutrients, detergents, contact with potential allergens (dust, animal hair, pollen), low air humidity, large temperature fluctuations.
Please describe the study group in more detail in terms of the above factors, as well as specify whether they were nomadic or sedentary families and whether high SCORAD values depended on the above conditions.
Moreover, please explain whether the severity of the disease process and the SCORAD, TEWL, skin moisture and pH values were dependent on the season. Certain climates can aggravate eczema, causing or worsening flare-ups. Climates with extreme hot or cold temperatures can dry out the skin. Those with extreme humidity can cause excessive sweating and associated skin irritation.
It widely known, that in AD patients skin hydration and epidermal integrity is important, and TEWL is increased, and skin moisture is reduced but we would like to know what are the differences in Mongolian population.
Therefore, I ask the authors to highlight new or original research results that could expand our knowledge of the living conditions of children in Mongolia and the specificity of the course of the disease in this population.
The work is properly written, has good statistical studies, clear tables and figures and up-to-date references. After some corrections could be published.
Author Response
Point-by-point response to the reviewer comments
Manuscript: jcm-3337268
Reviewer 3.
Atopic dermatitis (AD) is a chronic, recurrent dermatitis with periods of exacerbations and remissions. The disease is manifested by persistent itching, dryness and skin lichenification, what significantly contribute to the decrease of the quality of life of patients and their families. It is a widely known, common condition that usually appears in childhood, but adults also get sick. So, the topic is important.
Comment: In times when the way of life, climate and the amount of pollution are intensively changing, environmental factors play an important role in the severity of AD symptoms. The most common include: air pollution, exposure to stress, UV, certain nutrients, detergents, contact with potential allergens (dust, animal hair, pollen), low air humidity, large temperature fluctuations. Please describe the study group in more detail in terms of the above factors, as well as specify whether they were nomadic or sedentary families and whether high SCORAD values depended on the above conditions.
Response: Thank you for your insightful comment regarding environmental factors influencing AD severity. We recognize the significance of rising air pollution, limited sun exposure during winter and spring, and Mongolia’s low humidity due to its harsh climate as contributors to the chronic and relapsing nature of AD. In response, we have added relevant information to the Introduction (Lines 59-71, 77-90) to better contextualize these factors. Additionally, we incorporated evaluable data such as moisturizer using habits, housing and living area from the questionnaire, summarized in Table 1, and enhanced the regression analysis by including a model to account for potential confounding factors like socioeconomic status and other relevant variables. These updates, reflected in Tables 4A and 4B, provide a more comprehensive assessment of their impact on AD severity.
***.
Comment: Moreover, please explain whether the severity of the disease process and the SCORAD, TEWL, skin moisture and pH values were dependent on the season. Certain climates can aggravate eczema, causing or worsening flare-ups. Climates with extreme hot or cold temperatures can dry out the skin. Those with extreme humidity can cause excessive sweating and associated skin irritation. It widely known, that in AD patients skin hydration and epidermal integrity is important, and TEWL is increased, and skin moisture is reduced but we would like to know what are the differences in Mongolian population.
Response: We thank the reviewer for this insightful comment. While our study did not explicitly stratify data by season, we acknowledge that seasonal variations can significantly influence atopic dermatitis (AD) severity and skin barrier parameters such as SCORAD, TEWL, skin moisture, and pH. In Mongolia, the cold and dry weather during winter, combined with limited sun exposure in winter and spring, likely exacerbates AD by increasing TEWL and skin pH while decreasing skin moisture. These seasonal effects are well-documented in climates with extreme temperature fluctuations and low humidity. In response to the reviewer’s suggestion, we have incorporated additional information into the Introduction and Discussion sections, emphasizing potential contributors to altered TEWL and skin moisture in Mongolian children. These factors include Mongolia’s extreme climate, limited sun exposure, and genetic differences such as variations in FLG mutations. (Lines 59-71, 77-90, Figure 1) Furthermore, we have added a discussion of the study's strengths and limitations in the last paragraph of the Discussion section. (Lines 371-376)
***.
Comment: Therefore, I ask the authors to highlight new or original research results that could expand our knowledge of the living conditions of children in Mongolia and the specificity of the course of the disease in this population.
Response: As you highlighted, we adjusted for living conditions by categorizing participants into two groups: urban and countryside, as well as by region. However, we found no significant differences in disease severity across these categories (Table 1, 4A, 4B). We agree that finding a relationship between living conditions and AD severity would have been a notable result. Unfortunately, our analysis did not reveal such associations. It is important to note that Mongolia, with its harsh climate, may have pervasive environmental factors that could overshadow the impact of living conditions alone.
***.
The work is properly written, has good statistical studies, clear tables and figures and up-to-date references. After some corrections could be published.
Response: We thank the reviewer for their valuable feedback, which has helped strengthen the manuscript.
Reviewer 4 Report
Comments and Suggestions for Authors
Journal: JCM (ISSN 2077-0383)
Manuscript ID: jcm-3337268
Type: Article
Title: Skin physiological parameters and their association with severe atopic dermatitis in Mongolian children
Authors: Lkhamdari Batbileg , Sevjidmaa Baasanjav , Khosbayar Tulgaa , Oyuntugs Byambasukh , Khurelbaatar Naymdavaa , Enkhtur Yadamsuren * , Baasanjargal Biziya *
I appreciate the authors' effort in conducting this research. However, the research idea is not at all new. The paper distinguishes itself by the study cohort, which consists of patients from Mongolia.
The paper is carefully written and provides extensive information regarding the methodology, the division of patients into study groups, etc. However, the results were somewhat predictable.
Personally, I do not believe this paper is of broad interest to physicians.
I must admit that it is difficult for me to reject the article's publication, considering the effort invested in conducting the research.
The authors should add information that would differentiate this research from the many other papers published on this topic.
Author Response
Point-by-point response to the reviewer comments
Manuscript: jcm-3337268
Reviewer 4.
Comments:
I appreciate the authors' effort in conducting this research. However, the research idea is not at all new. The paper distinguishes itself by the study cohort, which consists of patients from Mongolia.
The paper is carefully written and provides extensive information regarding the methodology, the division of patients into study groups, etc. However, the results were somewhat predictable.
Personally, I do not believe this paper is of broad interest to physicians.
I must admit that it is difficult for me to reject the article's publication, considering the effort invested in conducting the research.
The authors should add information that would differentiate this research from the many other papers published on this topic.
Responses: We thank the reviewer for their thoughtful feedback and acknowledgment of the effort invested in this study. While we recognize that the research topic itself may not be entirely new, our study provides a unique perspective by focusing on Mongolian children with atopic dermatitis (AD), a population underrepresented in the current literature. Mongolia’s extreme climate, low humidity, and seasonal fluctuations present distinctive environmental challenges that may influence the skin barrier and disease severity. This study is the first to evaluate skin physiological parameters, such as TEWL, moisture, and pH, in this population.
To further differentiate our work:
- We have expanded Discussion to address the interplay of genetic, environmental, and skin-specific factors, which may explain the higher SCORAD index observed in our study compared to other populations (Lines 306, 315-326).
- Additional comparisons with data from other geographic regions and living conditions have been incorporated to highlight the relevance of our findings (Table 1, 4A, 4B).
- We have incorporated additional information on the internal and external factors specific to the Mongolian population to enhance the depth and relevance of the study. (Lines 59-71, 77-90).
- While some findings were anticipated, we have emphasized their significance in the context of Mongolia’s unique environmental and physiological conditions in the discussion section. This clarification not only underscores the importance of these results but also aids in advancing the understanding and management of AD in regions with similar environmental challenges (Lines 306, 370-378).
We thank Reviewer 4 for their thoughtful feedback and the opportunity to enhance our manuscript. In addition to your valuable comments, we have also carefully considered and incorporated feedback from three other reviewers. We hope these collective revisions have significantly strengthened our study.
Round 2
Reviewer 2 Report
Comments and Suggestions for Authors
The authors did take my suggestions into account and the manuscript is improved Well done !
Reviewer 4 Report
Comments and Suggestions for Authors
I have reviewed the changes and consider them appropriate and in line with the previous recommendations. I believe the article can be published in its current form.